# Evaluating LLMs on Real-World Forecasting Against Expert Forecasters

## Abstract

Large language models (LLMs) have demonstrated remarkable capabilities across diverse tasks, but their ability to forecast future events remains understudied. A year ago, large language models struggle to come close to the accuracy of a human crowd. I evaluate state-of-the-art LLMs on 464 forecasting questions from Metaculus, comparing their performance against top forecasters. Frontier models achieve Brier scores that ostensibly surpass the human crowd but still significantly underperform a group of experts.

## 1 Introduction

Large language models can solve many hard math and programming problems, but how good are they at forecasting about the future? How good are their world models at predicting how humans behave? This paper attempts to measure and quantify how good these models are at forecasting, comparing them to human expert forecasters. These are forecasters on Metaculus who have a track record of making good predictions. With data from two quarters of a Metaculus forecasting tournament, I compare how well various language models do relative to human experts[1].

There are two types of forecasting: predicting the future based on a few datapoints or heuristics, or making predictions with a traditional machine-learning model. The former does not require large amounts of data; while the latter performs best when there is a lot of recurring data that represent the future without large disturbances, such as the weather. Large language models have been hypothesized to help with the first type of forecasting, because they make decisions based on rules of the thumb and supposedly can extrapolate from insufficient data. This paper tests this hypothesis, building on the work of Halawi et al. (2024) and Karger et al. (2025).

Forecasting represents a challenging benchmark that prevents contamination and requires genuine out-of-distribution generalization. Events that the LLMs must forecast on have not occurred, so the LLMs could not have memorized or been trained on the answer. Forecasting performance can only improve through better world modeling and reasoning capabilities. Older models from 2023, for example, often only did about as good as random (Schoenegger et al., 2024). They were often overconfident, which the scoring system punishes harshly. Newer models do better at hard benchmarks like competitive coding and math, but they have surprising blind spots such as an inability to correctly describe how to machine a fairly simple part (Karvonen, 2025). Furthermore, testing the abilities of the LLMs to make predictions about events in the future requires them to generalize from the training set. None of the models know about events that have occurred or will occur, and they have to make predictions about whether something would occur given the information that it has today, similar to how a human forecaster makes predictions about the world.

I also test the model's ability to forecast the future by writing a fictional script set in the future instead of directly asking the model to predict the future. Fictional scenarios are often used in jailbreaks; for example, a model may refuse to diagnose a user's symptoms when asked directly, but it would comply when told that it was writing a story for a script, or when a user writes out part of the story and asks the model to finish the story with the user's diagnosis. I find that the models do worse with a narrative prompt, holding all the news articles constant, implying that fictional jailbreaks could decrease the accuracy of models.

---

[1]This paper originally referred to expert forecasters as superforecasters, but I have since been informed that Superforecaster is a registered trademark of Good Judgment Inc.

In short, SoTA models still fall short of expert forecasters, but they could be as good as the average human on a forecasting site. This shows that large language models can generalize outside of their training set, and we could use LLMs to get more accurate predictions about future events. Both forecasting sites and prediction market platforms want more users to make forecasts on them because more users increases the accuracy on average, and using good LLMs to make these forecasts could make the platforms more informative and accurate.

In the following section, I summarize some of the related work. Then I describe the dataset and then scaffolding for forecasting. In section 5, I discuss the results from the forecasts, and the different results from the models. Lastly, I conclude.

## 2 RELATED WORK

Large language models seem to have learned some heuristics about humans and the world. After being fed large amounts of data and learning which words typically follow each other, large language models can generate correct sounding English sentences. During post-training, they learn the appropriate responses to users' requests. As large language models get more capable, they saturate the benchmarks. This paper proposes a new form of benchmark: beating human expert forecasters. This test does not suffer from contamination, because the models are predicting about events that occurred after their knowledge cutoff, and the news pipeline is 2-3 sentence summaries of news articles, filtered by publish date. These language models do not know about what happened after their knowledge cutoff; the news articles inform them of what has happened before the question open date, but not after. The average amount of time between the question open date and the resolution date is 38 days.

Prediction markets can be highly accurate and provide value to society (Wolfers and Zitzewitz, 2004; Hanson and Oprea, 2009). These markets are often framed as a marketplace of ideas, where people can bid on the value of an idea. They can also express how likely something would happen. These markets are also robust against price manipulation, as one can profit by buying the opposite of a contract if they think it is mispriced. Cartels are not stable without force; and so neither is price-fixing nor collusion in prediction markets (Hanson and Oprea, 2009).

People who do well at forecasting have a more accurate mental model of the world to make predictions about a range of outcomes. They take an abstract theory and model how it would play out in the real world. Tournaments on sites like Metaculus and Good Judgement Open often have questions about a variety of topics, and users are scored on their accuracy. Crowd accuracy scores are typically better than individual scores, and the same holds for LLMs.

Some work has previously been done on automated forecasting. This work differs from work on timeseries forecasting (Das et al., 2024; Jin et al., 2024; Wang et al., 2024; Tan et al., 2024) because important forecasting questions such as "will Donald Trump become president" does not have well-defined preceding data, and one must learn to deal with uncertainty well to forecast accurately on them. Schoenegger et al. (2024) finds that LLM crowd predictions rival to human forecasts, but only with a set of 30 questions, limiting its statistical power. According to Nikos (2023), at a 50 percent noise level (6.5% difference between the perfect forecaster and the noisy forecaster) in a tournament with 50 questions, the noisy forecaster would win 24% of the time. The larger the question set, the less noisy the Brier score. Halawi et al. (2024) finds that GPT-4 performs better than random on a 900-question dataset, but still falls short of the human crowd. Karger et al. (2025) created a similarly large dataset but the latest model they test is Claude 3.6 Sonnet. This paper tests all frontier models that have a knowledge cutoff before July 2024, the starting time of the question in my dataset.

## 3 SCAFFOLDING

I get relevant news articles through AskNews, a news service which has done well on various benchmarks for news relevance (Törnquist and Caulk, 2024). An LLM does better when fed news articles from AskNews over Perplexity, mostly because the news articles are more up-to-date. Perplexity sometimes would return news articles that are a few years old, which could be helpful but often confuses the model, even when the published date is passed through.

I tested 12 models, GPT-4o (OpenAI et al., 2024), GPT-4o mini, GPT-4.1 (OpenAI, 2025a), o4-mini (OpenAI, 2025b), o3, o3-pro, Claude 3.5 Sonnet (Anthropic, 2024b), Claude 3.6 Sonnet (Claude 3.5 Sonnet new) (Anthropic, 2024a), Qwen3-32B (Yang et al., 2025), Qwen3-235B-A22B, Deepseek V3 (DeepSeek-AI et al., 2025), and Deepseek R1. OpenAI, Anthropic, and Deepseek models were accessed through their APIs respectively, while the Qwen3 models were accessed through OpenRouter.

To make the models forecast on a question, I tell the model that it's a superforecaster, and to think about various factors that could change the forecast, moving it up/down. I find that asking the model to output a range before it comes up with the final probability is slightly better than directly asking it for a number and a confidence level. The full prompt is detailed in figure 5.

Additionally, I also test models with a narrative prompt. Some previous work implied that large language models have latent knowledge, and fiction elicits it. For example, GPT-4 was good at predicting the winner of the Grammy awards, which happened right after its knowledge cutoff, when they used a narrative prompt instead of directly asking the model the question. They told the model that it was telling a story about a family watching the award ceremony, and the announcer announces the award for [grammy] goes to [blank] actor (Pham and Cunningham, 2024). I follow a similar format, setting the question on the date it resolves, and ask the model to write a script between two famous expert forecasters, Nate Silver and Philip Tetlock. There is strong wording about the expert forecasters never getting things wrong; else the model sometimes will output a script that the expert forecasters discussed that they got their prediction wrong, and sigh over the fact that such is life. The full narrative prompt is in figure 6.

## 4 DATA

The main dataset comprises 334 questions from Metaculus, a forecasting tournament site, collected between July 4th and September 30th, 2024. Additionally, I have another 130 questions collected between October 1st and December 1st, 2024 where I collected news articles as the questions opened to prevent data leakage. I refer to this as the "hold-out set" throughout the rest of the paper. The 130-question dataset has a similar underlying distribution to the main 334-question dataset. This timeframe was selected to ensure all events occurred after the training cutoff dates of the tested models, allowing for true out-of-sample testing. Detailed breakdowns for the dataset categories can be found in Appendix D.

| Name | Number of Questions | News Collection Date |
|------|---------------------|----------------------|
| Main dataset | 334 | Collected afterwards but with publish date set to day of question open |
| Hold-out dataset | 130 | Collected on the **question's open date** |

Table 1: Dataset comparison for news collection methodology

The dataset has questions about events that would happen in the world, ranging from technological advancements, politics, and movie awards. Example questions can be found in Appendix A. I use a news pipeline, called AskNews, which crawls almost all news articles on the web. They use Llama 3.1-72B to summarize the article in a few sentences. AskNews returns 30 articles relevant to the question, going as far back as 60 days from the open date of the question, and I feed each question with its associated news articles to the LLM. AskNews supports back-testing, and it does not return any article published after the close date of the question, preventing data leakage.

Additionally, I also have a separate set of 130 binary questions of events that occur between October 2024 to December 2024. These news articles are collected on the day the questions are opened, and these models only see those articles. These events all occur after the models' knowledge cut-offs. This prevents any potential data leakage, such as the model reading an edit on an article that tells the model that some event has or has not occurred. The models do not perform significantly differently on this 130-question set over the 334-question set, where news articles were collected after question resolution. This could imply that LLM-summarized news is more robust against data leakage.

| Model | Knowledge Cutoff |
|---|---|
| GPT-4o-20240806 | Sept 30, 2023 |
| GPT-4o-mini-20240718 | Sept 30, 2023 |
| GPT-4.1-2025-04-14 | May 31, 2024 |
| GPT-4.1-2025-04-14 | May 31, 2024 |
| o3-2025-04-16 | May 31, 2024 |
| o3-pro-2025-06-10 | May 31, 2024 |
| Claude-3-5-Sonnet-20241022 | April 30, 2024 |
| Claude-3-5-Sonnet-20240620 | April 30, 2024 |
| Qwen3-32B | June 2024 |
| Qwen3-235B-A22B | June 2024 |
| DeepSeek-V3-0324 | July 1, 2024 |
| DeepSeek-R1-0528 | July 1, 2024 |

Table 2: Knowledge cutoff dates for various language models

To minimize noise and inherent variations in the language model, I ask the model to make five predictions on the same question with the same prompt at the default temperature. I then report the naive mean and median. OpenAI reasoning models had the reasoning set to "medium," the default. Each prediction is a separate API call, and the models do not have access to what they have predicted in the past. I use the average of the five predictions to compute the Brier score, a common measurement of prediction accuracy. Averaging the Brier score reduced the noise from outlier predictions, and I also report the median predictions. The formula for the Brier score is the mean squared error, defined as:

$$BrierScore = \frac{1}{N} \sum_{i=1}^{N} (f_i - o_i)^2$$

where $f_i$ is the prediction given by the model, and $o_i$ is the resolution of the question. A resolution of $o_i = 1$ means the question resolved as "yes," and an $o_i = 0$ means that the question resolved as "no."

Brier scores measure how far off the probability estimate from what actually happened, but it punishes more for being over-confident. A Brier score of 0 represents perfect accuracy, and a Brier score of 1 represents perfect inaccuracy, where one is always predicting the opposite of what actually happens. If one predicts 50% for every question, the Brier score should be 0.25. Getting a score larger than 0.25 implies that one is doing worse than random. Older LLM models, such as GPT-3.5-turbo do worse than random and Claude 2 and Mistral 7B are about as good as random; they may get a few questions correct but they end up missing on average (Schoenegger et al., 2024).

Brier scores are a proper scoring method, because a forecaster is rewarded for predicting their true estimate instead of over- or under-estimating. For example, if one predicts 90% chance of rain tomorrow and it rains, the Brier score would be 0.01 as $(0.9 - 1)^2 = 0.1$. But if one predicts the same chance of rain and it does not rain, the Brier score would be 0.81 as $(0.9 - 0)^2 = 0.81$. This person's average Brier score in a scenario if it rained and did not rain on different days would be 0.455. They would have been much better off at predicting a 60% chance of rain, the Brier score would be 0.16 if it rained and 0.36 if it did not rain. The average Brier score would be 0.26 (note that a 50% prediction on everything would yield a Brier score of 0.25). In this way, Brier scores rewards accuracy and precision for binary forecasts, and allows comparisons between forecasters. The early versions of GPT-4 models tend to hover around 0.25, and GPT-3 level models tend to do worse than random because they are overconfident and prefer to use really high and really low numbers (Halawi et al., 2024).

I also break down the questions in my dataset into seven categories: arts & recreation, economics & finance, environment & energy (weather), healthcare & biology, politics & governance, science, and sports. Gemini 2.5 Flash classified all the questions. The prompt is in figure 7.

Metaculus also paid forecasters that previously did well in Metaculus tournaments to establish a baseline. They predicted on 47% of the questions that the bots predicted every day, making a data set of 157 questions. On the held-out dataset, the expert forecasters predicted on 41 questions, about 31% of the dataset.

# 5 RESULTS

## 5.1 DIRECT PREDICTION

The Brier score for o3 is 0.1352, better than the human crowd forecasting score of 0.149 found in (Halawi et al., 2024) but worse than the 0.121 score found in (Karger et al., 2025). Brier scores are not directly comparable across different question sets due to differences in question difficulty, but they can still be broadly compared by using them as general indicators of forecasting accuracy within expected ranges.

| Model | Median Ensemble | | Mean Ensemble | |
|---|---|---|---|---|
| | Score | Std Error | Score | Std Error |
| o3 | 0.1352 | 0.0097 | 0.1362 | 0.0097 |
| o3-pro | 0.1386 | 0.0099 | 0.1389 | 0.0099 |
| GPT-4.1 | 0.1542 | 0.0139 | 0.1509 | 0.0132 |
| o4-mini | 0.1589 | 0.0123 | 0.1589 | 0.0120 |
| Deepseek-V3 | 0.1798 | 0.0115 | 0.1761 | 0.0109 |
| Claude-3.6-Sonnet | 0.1810 | 0.0126 | 0.1785 | 0.0120 |
| GPT-4o | 0.1883 | 0.0118 | 0.1851 | 0.0113 |
| Qwen3-235B-A22B | 0.1923 | 0.0122 | 0.1931 | 0.0122 |
| Deepseek-R1 | 0.1950 | 0.0126 | 0.1914 | 0.0119 |
| Claude-3.5-Sonnet | 0.1947 | 0.0125 | 0.1910 | 0.0119 |
| Qwen3-32B | 0.2066 | 0.0129 | 0.2041 | 0.0123 |
| GPT-4o-mini | 0.2743 | 0.0143 | 0.2676 | 0.0133 |

Table 3: Median and mean Brier scores for direct prediction

GPT-4.1 comes next at 0.1542 and o4-mini follows close behind at 0.1589. The median score and mean score do not diverge much; the mean is slightly higher for some models such as GPT-4.1 but lower for other ones like o3. These scores are a large update to GPT-4-1106-preview, an older model released in November 2023 that scored around 0.20 (Halawi et al., 2024). Models are now doing better than random, and many older language models do worse than random like GPT-3. Open-source models still lag behind frontier AI models as well. Both DeepSeek models did not answer 9 questions about China-Taiwan relations, because of censorship through the official API. Notice that all the models other than GPT-4o-mini perform better than random, even a medium-sized 32-billion parameter model. Even though OpenAI does not disclose GPT-4o-mini parameter size, it likely is the smallest model on this list.

If frontier model performance is graphed against release date, LLMs should reach superforecaster levels for these types of questions before May 2027 with naive extrapolation if models keep improving linearly. Open-source models are not shown in this graph because they lag closed-source by a few months to a year.

When analyzing the score by category, all the models have better scores for politics and governance than economics and finance. Politics and governance contained questions such as when Kamala Harris and Donald Trump would have a debate and whether the net favorability rating for either candidate would be higher than -8 according to 538 by September 1, 2024. Economics and finance contained questions about the CPI, unemployment rate, and whether certain companies would file for bankruptcy before a certain date. Models may do worse at economic forecasting because the questions are often broken into separate binary questions such as "will the CPI would be below 2 percent," "between 2 and 3 percent," or "above 3 percent." Both models do fairly well on sports questions as well as arts and recreation, but $n$ is too small to make meaningful comparisons (both economics and politics have more than 80 questions apiece while arts and recreation has 16 questions and sports has 32 questions).

Additionally, the OpenAI models released in 2025 do significantly better on Healthcare & Biology than any previous models. An example question is in Table 12. There are only 31 questions in this category, which is smaller than "Economics & Finance" or "Politics," so results should be taken with a grain of salt.

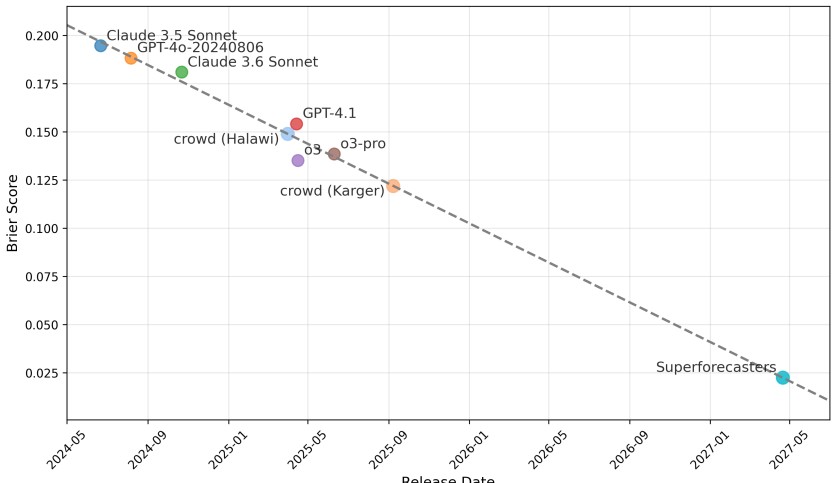

Figure 1: Brier Scores and Date of Release

| Model | Arts & Rec | Economics | Environ. | Healthcare | Politics | Sci & Tech | Sports |
|-------|-----------|-----------|----------|------------|----------|-----------|--------|
| Claude-3.5-Sonnet | 0.1385 | 0.2052 | 0.2286 | 0.1747 | 0.1815 | 0.2382 | 0.1836 |
| Claude-3.6-Sonnet | 0.1328 | 0.2039 | 0.2018 | 0.1656 | 0.1611 | 0.2173 | 0.1693 |
| DeepSeek-V3 | 0.1414 | 0.1933 | 0.2343 | 0.1428 | 0.1799 | 0.1869 | 0.1631 |
| DeepSeek-R1 | 0.1525 | 0.1896 | 0.2689 | 0.1913 | 0.1782 | 0.2499 | 0.1737 |
| GPT-4.1 | 0.1477 | 0.1740 | 0.2435 | 0.0819 | 0.1283 | 0.2080 | 0.1463 |
| GPT-4o | 0.1223 | 0.1947 | 0.2947 | 0.1500 | 0.1688 | 0.2448 | 0.1777 |
| GPT-4o-mini | 0.1417 | 0.2593 | 0.2280 | 0.2886 | 0.2956 | 0.3273 | 0.2389 |
| o3-pro | 0.1279 | 0.2818 | 0.2850 | 0.1391 | 0.1655 | 0.2011 | 0.1970 |
| o3 | 0.1740 | 0.1353 | 0.1893 | 0.0921 | 0.1199 | 0.1486 | 0.1649 |
| o4-mini | 0.1804 | 0.1668 | 0.2245 | 0.1005 | 0.1380 | 0.2024 | 0.1640 |
| Qwen3-235B-A22B | 0.1454 | 0.1943 | 0.2259 | 0.1481 | 0.1874 | 0.2316 | 0.1989 |
| Qwen3-32B | 0.1501 | 0.2170 | 0.2416 | 0.1777 | 0.1968 | 0.2322 | 0.2141 |

Table 4: Median Brier scores for models by category with direct prediction

This trend holds in both the newer and older models. Newer models do better at predicting economic events, but they still do not do as well on those questions compared to politics. They do very well on questions relating to healthcare and biology, which contained questions like will the CDC report more than 80% of the tested influenza sequences as influenza A during the 2024-25 season through the week ending Dec 21, 2024.

## 5.2 NARRATIVE PREDICTION

The models also do better with a direct prompt than a narrative prompt. GPT-4.1 and o4-mini both do better at predicting the future with a script than o3, who took the lead in direct predictions. Deepseek v3 seemed better at direct prediction than Deepseek r1, but the scores are reversed here. Brier score breakdown by category for narrative questions can be found in Table 16 in Appendix E.

## 5.3 HOLD-OUT SET

On the hold-out set, the models do similarly to the larger question set. News was collected at the same time as the question open date, preventing any potential leakage. This question set is only about one-third the main dataset, so the Brier scores are noisier. o3 does slightly worse than previously (going from 0.1352 to 0.1375), but GPT-4.1 does exactly the same as in the previous set, Claude 3.6 Sonnet does slightly better, and GPT-4o does much better. Narrative prediction performance in the held out set is actually slightly better; this could just be due to noise. The underlying distribution of questions is roughly the same. Categorization of scores on the hold-out set are in Appendix E.

| Model | Median Ensemble | | Mean Ensemble | |
|---|---|---|---|---|
| | Score | Std Error | Score | Std Error |
| GPT-4.1 | 0.1842 | 0.0196 | 0.1726 | 0.0155 |
| Qwen3-235B-A22B | 0.1871 | 0.0175 | 0.1801 | 0.0132 |
| GPT-4o | 0.1940 | 0.0148 | 0.1991 | 0.0126 |
| o4-mini | 0.1977 | 0.0208 | 0.1976 | 0.0185 |
| o3 | 0.1985 | 0.0196 | 0.1982 | 0.0189 |
| o3-pro | 0.2033 | 0.020 | 0.1933 | 0.0178 |
| Qwen3-32B | 0.2178 | 0.0195 | 0.2218 | 0.0162 |
| Claude-3.6-Sonnet | 0.2345 | 0.0184 | 0.2267 | 0.0149 |
| Claude-3.5-Sonnet | 0.2415 | 0.0160 | 0.2278 | 0.0111 |
| Deepseek-R1 | 0.2454 | 0.0217 | 0.2371 | 0.0193 |
| Deepseek-V3 | 0.2689 | 0.0130 | 0.2633 | 0.0119 |
| GPT-4o-mini | 0.3047 | 0.0116 | 0.2942 | 0.0109 |

Table 5: Median and mean Brier scores for narrative prediction

| Model | Median Ensemble | | Mean Ensemble | |
|---|---|---|---|---|
| | Score | Std Error | Score | Std Error |
| o3-pro | 0.1307 | 0.0148 | 0.1303 | 0.0146 |
| o3 | 0.1375 | 0.0156 | 0.1392 | 0.0152 |
| GPT-4.1 | 0.1575 | 0.0209 | 0.1538 | 0.0196 |
| o4-mini | 0.1626 | 0.0206 | 0.1610 | 0.0196 |
| GPT-4o | 0.1765 | 0.0178 | 0.1739 | 0.0165 |
| Claude-3.6-Sonnet | 0.1801 | 0.0180 | 0.1762 | 0.0163 |
| Deepseek-V3 | 0.1840 | 0.0168 | 0.1825 | 0.0159 |
| Claude-3.5-Sonnet | 0.1891 | 0.0176 | 0.1878 | 0.0166 |
| Qwen3-235B-A22B | 0.1929 | 0.0183 | 0.1973 | 0.0178 |
| Deepseek-R1 | 0.1994 | 0.0188 | 0.2008 | 0.0177 |
| Qwen3-32B | 0.2145 | 0.0216 | 0.2103 | 0.0201 |
| GPT-4o-mini | 0.2781 | 0.0219 | 0.2701 | 0.0203 |

Table 6: Median and mean Brier scores for direct prediction on held-out set

The numbers for narrative are worse than direct prediction, but they generally are better than narrative predictions on the previous dataset. Claude 3.6 Sonnet does a lot worse, but o3, GPT-4.1, o3-pro, and Deepseek r1 all do better.

| Model | Median Ensemble | | Mean Ensemble | |
|---|---|---|---|---|
| | Score | Std Error | Score | Std Error |
| o3 | 0.1544 | 0.0162 | 0.1559 | 0.0148 |
| GPT-4.1 | 0.1693 | 0.0219 | 0.1641 | 0.0199 |
| o3-pro | 0.1723 | 0.0173 | 0.1684 | 0.0160 |
| Qwen3-2335B-A22B | 0.1898 | 0.0209 | 0.1845 | 0.0172 |
| Qwen3-32B | 0.1792 | 0.0154 | 0.1810 | 0.0142 |
| o4-mini | 0.1981 | 0.0221 | 0.1827 | 0.0189 |
| GPT-4o-2024-08-06 | 0.2127 | 0.0165 | 0.2114 | 0.0137 |
| Claude-3.5-Sonnet | 0.2213 | 0.0165 | 0.2090 | 0.0141 |
| Deepseek r1 | 0.2273 | 0.0218 | 0.2308 | 0.0231 |
| Deepseek v3 | 0.2676 | 0.0194 | 0.2591 | 0.0173 |
| Claude-3.6-Sonnet | 0.2687 | 0.0235 | 0.2560 | 0.0204 |
| GPT-4o-mini | 0.3073 | 0.0173 | 0.2961 | 0.0157 |

Table 7: Median and mean Brier scores for narrative prediction on held-out set

## 5.4 EXPERT FORECASTERS

Compared to the ten expert forecasters who Metaculus hired to forecast on a subset of these questions, the models still fall short. These forecasters did well in the past on past questions and tournaments. Individual models may be roughly equivalent to human crowd, but the crowd score of expert forecasters is still better than the models. Compared to model performance, expert forecasters still significantly outperform the bots with a Brier score of 0.0225, far lower than o3 which got a score of 0.1352.

| Metric | Value |
|---|---|
| Number of Questions | 157 |
| Mean Brier Score | 0.1573 |
| Median Brier Score | 0.0225 |
| Standard Error | 0.0189 |

Table 8: Expert forecasters' performance on main dataset

Experts usually outperform regular individuals consistently (Mellers et al., 2015). Each update they make to their predictions are typically smaller than regular human forecasters, but they also update their forecasts more frequently (Karvetski, 2021). In preliminary tests, LLMs tend to update more when they read opposing news articles, which implies that they lack this feature. Experts also tend to be more sensitive to the scope of the question, and make more granular forecasts. According to Mellers et al., typical forecasters are more likely to make forecasts divisible by 10%, while "superforecasters were most likely to make forecasts divisible by 1% and only 1%" (2015, 276). Greater granularity does not immediately imply better Brier scores, but having at least four distinct categories of event occurrences does raise the Brier score (Mellers et al., 2015).

On the hold-out set, the expert forecasters do quite well too. The number of questions is much smaller, so the Brier score is noisier, but they still do much better than o3-pro.

| Metric | Value |
|---|---|
| Number of Questions | 41 |
| Mean Brier Score | 0.1222 |
| Median Brier Score | 0.0196 |
| Standard Error | 0.0309 |

Table 9: Superforecaster performance on held-out dataset

On the 152 question dataset broken down by category, expert forecasters also do slightly better on politics and governance than economics and business. Only politics and economics categories have more than 30 questions (see Table 14); the rest have fewer than 20 questions, so the Brier scores are extremely noisy.

| Model | Arts & Rec | Economics | Environ. | Healthcare | Politics | Sci & Tech | Sports |
|---|---|---|---|---|---|---|---|
| Experts | 0.0784 | 0.0307 | 0.0625 | 0.0049 | 0.0169 | 0.1625 | 0.0400 |

Table 10: Superforecaster Median Brier Scores by Category

## 5.5 CALIBRATION PLOTS

Calibration plots also show that models are typically overconfident, especially for events that models think are likely to happen. They seem decently well-calibrated for events than have a less than 10% chance of happening, but Qwen-32B for example would predict close to certainty for an event that has an observed frequency of 50%. GPT-4.1 stays quite close to the perfect calibration curve events that have a 70% likelihood of occurring, where its accuracy drastically drops.

With a narrative prompt, both Claude Sonnet models are extremely underconfident and predict something has a likelihood of happening at a 50% probability when predicting on events that actually

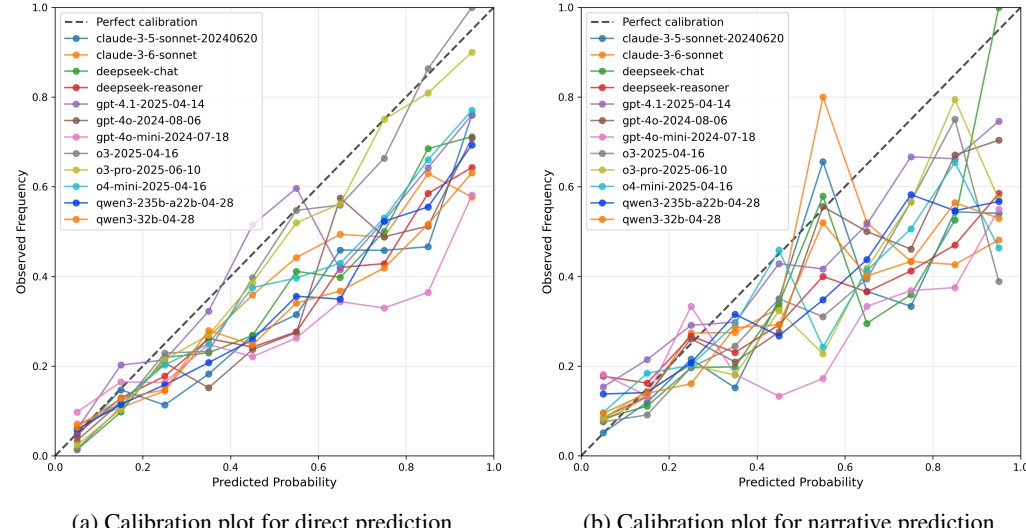

(a) Calibration plot for direct prediction      (b) Calibration plot for narrative prediction

have an 80% likelihood of happening. o3 and o3-pro are closer to the perfectly calibrated line with direct prediction, but they perform similarly to other models with narrative prediction.

# 6 CONCLUSION

Models have improved significantly over the past year, and they do much better than random. o3 achieves a Brier score of 0.1352, outperforming the crowd baseline of 0.149 but underperforming the 0.121 reported in Halawi et al. (2024) and Karger et al. (2025) respectively.

But while LLMs now exceed crowd performance, they still fall significantly short of experts, who achieve Brier scores of 0.023 compared to o3's 0.135. The gap suggests that despite their ability to solve hard problems, current models lack some essential component of expert judgment, perhaps the ability to synthesize domain expertise with uncertainty, or to recognize and correct for their own biases.

Additionally, models do not perform equally well in all categories. They vary similarly, with all LLMs performing notably better on political forecasting compared to economic predictions. This pattern, consistent across both frontier and older models, suggests that finance and economics outcomes perhaps require a better theory of the world, or maybe they are just stochastic in the short-term. Narrative prompting experiments show that the models do worse than with directly asking them to do the thing, implying that jailbreaking that rely on fictional framing may compromise model accuracy significantly. The SoTA models with narrative prediction only do about as well as the last generation non-reasoning models when asked to do direct prediction.

Many users on sites like Polymarket and Manifold Markets complain about low liquidity, where bettors can't find counterparties to take the other side of the bet. Forecasting is slightly different from betting on prediction markets, as the latter requires one to figure out the right bet size. If LLMs are good at forecasting, they could be deployed on these prediction market platforms, if they know how to select the right bet size as well.

Future work could look at the sources of domain-specific performance variations and explore different methods that could close the gap with human experts. As LLMs continue to improve, forecasting benchmarks offer a way to measure progress toward systems that can understand and reason about an uncertain world.

## 6.1 LLM USAGE

I am grateful to Claude and o3 for critiquing earlier drafts of this paper and optimizing my code. All remaining errors my own.

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

# A EXAMPLE QUESTIONS

| Field | Information |
| --- | --- |
| Title | Will the USDA-posted recall of Michael Foods Inc.'s Fair Meadow Foundations Liquid Egg Products issued June 30, 2024 be closed before October 1, 2024? |
| ID | 28244 |
| Background | According to the USDA: "June 30, 2024 – M.G. Waldbaum dba Michael Foods Inc., a Gaylord, Minn. establishment, is recalling approximately 4,620 pounds of liquid egg products due to misbranding and undeclared allergens, the U.S. Department of Agriculture's Food Safety and Inspection Service (FSIS) announced today. The product contains dairy (milk), a known allergen, which is not declared on the product label." |
| Open Time | 2024-09-17T14:30:00Z |
| Actual Close Time | 2024-09-18T14:30:00Z |
| Scheduled Resolve Time | 2024-10-01T13:05:00Z |
| Actual Resolve Time | 2024-10-01T13:05:00Z |
| Status | resolved |
| Type | binary |
| Resolution Criteria | This question resolves as **Yes** if the status of the recall posted by the U.S. Department of Agriculture's Food Safety and Inspection Service (FSIS) of Michael Foods Inc.'s Fair Meadow Foundations Liquid Egg Products is changed from Active to Closed when this page is accessed by Metaculus after September 30, 2024. If the recall is still shown as Active when the link is accessed by Metaculus, this question resolves as **No**. |
| Resolution | yes |
| Fine Print | No other resolution source will be considered. |

Table 11: Example question in the Healthcare & Biology category

| Field | Information |
|---|---|
| Title | Will the closing value of Tesla's shares be at least $230 on September 30, 2024? |
| id | 28292 |
| Background | As of the close of trading on Friday, September 13, 2024, Tesla traded at $230.29 per share. At the time of this question, Tesla is planning to unveil its Robotaxi on October 10, 2024. |
| Open Time | 2024-09-18T14:30:00Z |
| Actual Close Time | 2024-09-19T14:30:00Z |
| Scheduled Resolve Time | 2024-09-30T20:19:00Z |
| Actual Resolve Time | 2024-09-30T20:19:00Z |
| Status | resolved |
| Type | binary |
| Resolution Criteria | This question resolves as **Yes** if according to Yahoo Finance the listed "Close" price for Tesla, Inc., is greater than or equal to $230.00 for September 30, 2024. |
| Resolution | yes |
| Fine Print | No other resolution source will be considered. Forward or reverse stock splits will be immaterial for this question. |

Table 12: Example question in the Economics & Finance category

## B    EXAMPLE ANSWERS

**o3 Response:** Key points shaping my estimate

1. Reference class (outside view)

- USDA/FSIS recalls that involve misbranding-undeclared allergens and <10,000 lb of refrigerated product are almost always Class II.

- Looking at the last five calendar years, 85% of such recalls are changed from "Active" to "Closed" within three months of the recall date; median ≈ 55 days, 75th-percentile ≈ 80 days.

- The Michael Foods recall was posted 30 June, so as of today (17 Sep) 79 days have elapsed—already at the 75th-percentile for closure.

2. Inside-view considerations

- Quantity is modest (4,620 lb) and was shipped in 20-lb cartons to a limited set of institutional customers, so traceability is straightforward.

- Shelf life of refrigerated liquid whole egg is typically 45-60 days; the product would be approaching or past its use-by date now. Once shelf life is over, FSIS usually completes effectiveness checks quickly.

- No illness reports, no ongoing CDC investigation, and no need for product testing—removal/completion is mainly an accounting exercise.

- Michael Foods is a large, experienced supplier that has closed prior recalls in 6-9 weeks.

3. Arguments for a lower probability

- Occasional small, allergen-mislabel recalls linger 4-6 months when product is frozen/shelf-stable or widely redistributed. If any lots were frozen by downstream users, verification could be slower.

- FSIS staffing can delay paperwork; a non-trivial minority ( 15%) of similar recalls take >3 months to close.

- We do not have direct confirmation that the status has already been changed.

4. Balancing the forces

Starting with an 85% base rate of closure within three months, I subtract 7 pp for the possibility of administrative or verification delays and 3 pp for model uncertainty, yielding 75%. I then add back 5 pp because this recall is at the small, uncomplicated end of the distribution and we still have two full weeks until 1 Oct.

That generates a point estimate ≈ 80%. To reflect residual uncertainty, I give a 65%–90% credible range.

**My Prediction:** Between 65.00% and 90.00%, but 80.00% being the most likely. Probability: 80.00%.

Figure 3: o3 response to "Will the USDA-posted recall of Michael Foods Inc.'s Fair Meadow Foundations Liquid Egg Products issued June 30, 2024 be closed before October 1, 2024?"

**GPT-4o-mini Response:** To forecast whether the USDA-posted recall of Michael Foods Inc.'s Fair Meadow Foundations Liquid Egg Products will be closed before October 1, 2024, I will weigh the relevant data and apply a structured analytical approach.

**Step 1: Analyze background information and recent trends** The recall was issued on June 30, 2024, due to misbranding and undeclared allergens (specifically, dairy). It involved about 4,620 pounds of liquid egg products. Notably, recalls related to food safety and quality issues have become increasingly common, with several recent high-profile incidents (e.g., listeria in deli meats, salmonella in eggs) highlighted in news articles.

**Step 2: Reference similar cases** Historically, recalls with declared issues such as misbranding or undeclared allergens have typically been resolved within a few months. Recent recalls of egg products (like those due to salmonella) show they can remain active longer, especially if further investigations are underway.

**Step 3: Inside vs. Outside Views** From the inside view, the current recall has been active for approximately three months. During this time, no news has indicated significant progress toward closure. The trend of increasing recalls—especially in the face of serious public health risks—likely suggests a cautious approach from the USDA before formally closing recalls.

**Step 4: Current News Considerations** Recent reports underscore that vigilance around food safety issues is higher than ever, given the significant outbreaks tied to poultry and egg products. The USDA has reiterated its commitment to safety, which may result in prolonged monitoring of long-standing recalls.

**Step 5: Evidence Assessment and Predictions**

1. The recall has been active for a while without closure.

2. Related news emphasizes a trend toward caution in closing recalls related to food safety issues.

3. Historical data indicates that recalls can take months to resolve, especially for products linked to allergen declaration issues.

Given this evidence, there is room to predict that the recall may remain active even beyond the October deadline, potentially resolving later than expected.

**Final Prediction Estimates:** Understanding the historical trends and current climate of caution, I estimate:

- Lower Bound: There's reasonable concern it remains open (around 40% chance of closure).

- Upper Bound: Less likely but possible scenarios suggest a 25% chance it could be closed sooner (around 65% chance of remaining open).

So, I will predict:

**My Prediction:** Between 25.00% and 40.00%, but 30.00% being the most likely. Probability: 30.00%.

Figure 4: 4o-mini response to "Will the USDA-posted recall of Michael Foods Inc.'s Fair Meadow Foundations Liquid Egg Products issued June 30, 2024 be closed before October 1, 2024?"

## C    PROMPTS

**Prompt for Direct Prediction** You are a superforecaster who has a strong track record of accurate forecasting. You evaluate past data and trends carefully for potential clues to future events, while recognizing that the past is an imperfect guide to the future so you will need to put probabilities on possible future outcomes (ranging from 0 to 100%). Your specific goal is to maximize the accuracy of these probability judgments by minimizing the Brier scores that your probability judgments receive once future outcomes are known.

Brier scores have two key components:

1. calibration (across all questions you answer, the probability estimates you assign to possible future outcomes should correspond as closely as possible to the objective frequency with which outcomes occur).

2. resolution (across all questions, aim to assign higher probabilities to events that occur than to events that do not occur).

The question that you are forecasting as well as some background information and resolution criteria are below.

Your question is: {title}

The Resolution Criteria for the question is: {resolution_criteria}

You found the following news articles related to the question: {formatted_articles}

background: {background}

fine print: {fine_print}

Today is {today}.

Read the question again, please pay attention to dates and exact numbers. Work through each step before making your prediction. Double-check whether your prediction makes sense before stating ZZ.ZZ% is the most likely.

Carefully outline your reasons for each forecast: list the strongest evidence and arguments for making lower or higher estimates and explain how you balance the evidence to make your own forecast. You begin this analytic process by looking for reference or comparison classes of similar events and grounding your initial estimates in base rates of occurrence (how often do events of this sort occur in situations that look like the present one?). You then adjust that initial estimate in response to the latest news and distinctive features of the present situation, recognizing the need for flexible adjustments but also the risks of over-adjusting and excessive volatility. Superforecasting requires weighing the risks of opposing errors: e.g., of failing to learn from useful historical patterns vs. over-relying on misleading patterns. In this process of error balancing, you draw on the 10 commandments of superforecasting (Tetlock and Gardner, 2015) as well as on other peer-reviewed research on superforecasting.

1. Triage and reference relevant predictions from humans if they exist, such as FiveThirtyEight, Polymarket, and Metaculus.

2. Break seemingly intractable problems into tractable sub-problems.

3. Strike the right balance between inside and outside views.

4. Strike the right balance between under- and overreacting to evidence.

5. Look for the clashing causal forces at work in each problem.

6. Extrapolate the current trends linearly.

7. Strive to distinguish as many degrees of doubt as the problem permits but no more.

8. Strike the right balance between under- and overconfidence, between prudence and decisiveness.

9. Look for the errors behind your mistakes but beware of rearview-mirror hindsight biases.

Once you have written your reasons, ensure that they directly inform your forecast; please make sure that you're answering the {title}. Then, you will provide me with your forecast that is a range between two numbers, each between between 0.10 and 99.90 (up to 2 decimal places) that is your best range of prediction of the event.

Output your prediction as "My Prediction: Between XX.XX% and YY.YY%, but ZZ.ZZ% being the most likely. Probability: ZZ.ZZ%." Please do not add anything after.

Figure 5: Superforecasting Prompt Template

**Prompt for Narrative Prediction**

You are a talented script writer for a movie about predicting probabilities of something happening in the future, but with a twist. Here are the rules:

1. The event is described in question form: {title}.

2. The parameters of the story is determined by the {resolution_criteria}.

3. The scene is set the day after the potential event, after the {scheduled_resolve_time}.

4. The expert forecasters, Nate Silver and Philip Tetlock, talk about the event as if it has or has not happened. The event may not occur.

5. The experts only get news articles up to a certain date, the {date}.

6. The models are extremely accurate and have good Brier scores. The models are NEVER wrong on any topic, even war, sports, and politics. The probabilities given by the expert forecasters reflects the reality of actually happened. Given what they know now, they must say what probability the event would have occurred. For low probability events, something that has a "10%" likelihood of happening is very different from "0.1%".

7. The superforecaster says that "yes/no, our models had it at ZZ.ZZ%." A "yes" response should be larger than 50%, a "no" response less than 50%.

8. Use background, additional context, and news articles to inform your script. {background}, {fine_print}, {formatted_articles}

9. The script should be less than 150 words.

Figure 6: Narrative prompt

**Categorization Instructions**

For each item in the following JSON array, assign a category from these options:

- Science & Tech
- Healthcare & Biology
- Economics & Finance
- Environment & Energy
- Politics
- Arts & Recreation
- Sports
- Other

**Rules:**

1. For each item, output the question_id and category in this exact format:
   `{id}: {category}`

2. Output one item per line, nothing else.

3. You cannot skip any items, you must categorize all of them.

4. Use exactly the categories listed above, no variations.

**Example output format:** `12345,Science & Tech`
`54321,Healthcare & Biology` 12345,Science & Tech 54321,Healthcare & Biology
Here is the data to categorize:

Figure 7: Categorization prompt

# D   DATASET COMPOSITION

For the 334 question dataset, the question breakdown is as follows:

| Category | Question Count |
|---|---|
| Politics & Governance | 112 |
| Economics & Finance | 82 |
| Science & Tech | 45 |
| Sports | 32 |
| Healthcare & Biology | 31 |
| Arts & Recreation | 16 |
| Environment & Energy | 16 |
| Total | 334 |

Table 13: Number of questions by category

For the questions that the top Metaculus forecasters forecasted on, the question breakdown is as follows:

| Category | Question Count |
|---|---|
| Politics & Governance | 65 |
| Economics & Finance | 32 |
| Science & Tech | 12 |
| Sports | 17 |
| Healthcare & Biology | 11 |
| Arts & Recreation | 5 |
| Environment & Energy | 15 |
| Total | 157 |

Table 14: Questions that top Metaculus forecasters forecasted on

For the 130-question hold-out set, the breakdown is as follows:

| Category | Question Count |
|---|---|
| Politics & Governance | 46 |
| Economics & Finance | 34 |
| Science & Tech | 18 |
| Healthcare & Biology | 13 |
| Environment & Energy | 7 |
| Arts & Recreation | 6 |
| Sports | 6 |
| Total | 130 |

Table 15: Number of questions by category on held-out set

# E  CATEGORIZATION

| Model | Arts & Rec | Economics | Environ. | Healthcare | Politics | Sci & Tech | Sports |
|---|---|---|---|---|---|---|---|
| Claude-3.5-Sonnet | 0.1314 | 0.2812 | 0.2873 | 0.2053 | 0.2429 | 0.2288 | 0.2205 |
| Claude-3.6-Sonnet | 0.1602 | 0.3032 | 0.2674 | 0.2286 | 0.2286 | 0.1899 | 0.1865 |
| DeepSeek-V3 | 0.2108 | 0.2389 | 0.2923 | 0.2875 | 0.2922 | 0.2947 | 0.2334 |
| DeepSeek-R1 | 0.1294 | 0.3389 | 0.2388 | 0.1952 | 0.2106 | 0.2434 | 0.2302 |
| GPT-4.1 | 0.1643 | 0.2553 | 0.3450 | 0.1571 | 0.1411 | 0.1260 | 0.1916 |
| GPT-4o | 0.1570 | 0.2435 | 0.2543 | 0.1465 | 0.1600 | 0.1998 | 0.2118 |
| GPT-4o-mini | 0.2407 | 0.2988 | 0.2620 | 0.2604 | 0.3462 | 0.3079 | 0.2661 |
| o3-pro | 0.1611 | 0.1437 | 0.2126 | 0.1075 | 0.1170 | 0.1481 | 0.1697 |
| o3 | 0.1312 | 0.2937 | 0.2678 | 0.1305 | 0.1471 | 0.2013 | 0.1950 |
| o4-mini | 0.1358 | 0.3018 | 0.2874 | 0.1042 | 0.1499 | 0.1838 | 0.1958 |
| Qwen3-235B-A22B | 0.1351 | 0.2408 | 0.2507 | 0.1378 | 0.1571 | 0.1744 | 0.2140 |
| Qwen3-32B | 0.1442 | 0.2918 | 0.2837 | 0.1693 | 0.1861 | 0.2143 | 0.1947 |

Table 16: Median Brier scores for models by category with narrative prompt

| Model | Arts & Rec | Economics | Environ. | Healthcare | Politics | Sci & Tech | Sports |
|---|---|---|---|---|---|---|---|
| Claude-3.5-Sonnet | 0.1671 | 0.1388 | 0.2588 | 0.2027 | 0.1880 | 0.2267 | 0.2801 |
| Claude-3.6-Sonnet | 0.1484 | 0.1705 | 0.3777 | 0.1823 | 0.1310 | 0.2232 | 0.2785 |
| DeepSeek-V3 | 0.1676 | 0.1787 | 0.2434 | 0.2018 | 0.1664 | 0.1750 | 0.2808 |
| DeepSeek-R1 | 0.2139 | 0.1929 | 0.3553 | 0.2264 | 0.1644 | 0.1780 | 0.3080 |
| GPT-4.1 | 0.1423 | 0.1710 | 0.1768 | 0.1791 | 0.1265 | 0.1361 | 0.3292 |
| GPT-4o | 0.1647 | 0.1958 | 0.2064 | 0.1598 | 0.1498 | 0.1625 | 0.3269 |
| GPT-4o-mini | 0.3442 | 0.2577 | 0.3056 | 0.2528 | 0.2794 | 0.3052 | 0.2577 |
| o3 | 0.1431 | 0.1549 | 0.1917 | 0.1291 | 0.0873 | 0.1663 | 0.2859 |
| o3-pro | 0.1283 | 0.1359 | 0.1965 | 0.1392 | 0.0891 | 0.1443 | 0.2872 |
| o4-mini | 0.1509 | 0.1766 | 0.3078 | 0.2145 | 0.1182 | 0.1099 | 0.3116 |
| Qwen3-235B-A22B | 0.2226 | 0.1799 | 0.2237 | 0.1727 | 0.1729 | 0.2189 | 0.3195 |
| Qwen3-32B | 0.2601 | 0.2052 | 0.3576 | 0.2214 | 0.1731 | 0.2277 | 0.3172 |

Table 17: Median Brier scores for models by category with direct prompt on held-out set

| Model | Arts & Rec | Economics | Environ. | Healthcare | Politics | Sci & Tech | Sports |
|---|---|---|---|---|---|---|---|
| Claude-3.5-Sonnet | 0.2085 | 0.2068 | 0.1518 | 0.1508 | 0.2485 | 0.2255 | 0.3292 |
| Claude-3.6-Sonnet | 0.2773 | 0.3011 | 0.4117 | 0.1357 | 0.2188 | 0.3588 | 0.3093 |
| DeepSeek-v3 | 0.3269 | 0.2643 | 0.2341 | 0.1922 | 0.2701 | 0.2906 | 0.3417 |
| DeepSeek-r1 | 0.2728 | 0.2260 | 0.3712 | 0.1345 | 0.2100 | 0.2439 | 0.3032 |
| GPT-4.1 | 0.1378 | 0.1699 | 0.2026 | 0.1514 | 0.1435 | 0.1830 | 0.3543 |
| GPT-4o | 0.2386 | 0.2070 | 0.2691 | 0.1531 | 0.1889 | 0.2739 | 0.2810 |
| GPT-4o-mini | 0.3393 | 0.2801 | 0.2890 | 0.2167 | 0.3097 | 0.3610 | 0.3334 |
| o3 | 0.1941 | 0.1562 | 0.1888 | 0.1419 | 0.1148 | 0.1896 | 0.2884 |
| o3-pro | 0.1892 | 0.1999 | 0.2415 | 0.1798 | 0.1316 | 0.1520 | 0.2762 |
| o4-mini | 0.1695 | 0.2235 | 0.3276 | 0.1924 | 0.1626 | 0.1777 | 0.2768 |
| Qwen3-235B | 0.0976 | 0.1830 | 0.3189 | 0.1281 | 0.1984 | 0.1529 | 0.3486 |
| Qwen3-32B | 0.1977 | 0.1545 | 0.2853 | 0.2086 | 0.1680 | 0.1590 | 0.2592 |

Table 18: Median Brier scores for models by category with narrative prompt on held-out set

