# OpenReview forum: "Evaluating LLMs on Real-World Forecasting Against Expert Forecasters"
_ICLR.cc/2026/Conference — Submitted to ICLR 2026_

### Official Review · Reviewer_nVva · 2025-10-30

**Soundness:** 1
**Presentation:** 1
**Contribution:** 2
**Rating:** 2
**Confidence:** 4

**Summary:**

The paper studies the current state of LLMs on the forecasting task.

**Strengths:**

- Some results/analysis may be interesting to the community.
- Study on a recently new application of LLMs.

**Weaknesses:**

The paper reads like a poorly written project report. Specifically, section 3 severely lacks clarity.

**Questions:**

- "I" is used throughout the paper, but the author should use "We"
- The text is riddled with language errors, and the writing is very subpar.
- Citations do not seem to follow the standard ICLR style. This needs to be fixed.
- Many figures aren't referenced in the text.

---

> ### Author Response · Authors · 2025-11-21
>
> We thank the reviewer for pointing out issues in the presentation of the work and we will happily improve the writing based on the reviewer’s comments. Given that the reviewer does not raise any substantial concerns; in particular, no issues in understanding the technical content of the paper, we do not see how these relatively minor presentation issues (which we will address in the camera ready version of the paper) warrant a rejection.

---

### Official Review · Reviewer_ih4C · 2025-10-30

**Soundness:** 2
**Presentation:** 3
**Contribution:** 1
**Rating:** 2
**Confidence:** 4

**Summary:**

The paper considers forecasting as an out-of-distribution test for LLMs by evaluating them on events that had not yet occurred at the time of model training. The study collections ~400 questions from Metaculus and tests multiple frontier models using a “superforecaster” style prompt. As its main result, the best model (o3) achieves ≈0.135 Brier score. This is better than historical human-crowd baselines from prior work on different question sets but still underperforms expert forecasters. It also shows that narrative prompting degrades performance relative to direct forecasting prompts, suggesting that fictional framings can hurt probabilistic calibration.

**Strengths:**

The paper is overall well written with a clean main result.  The measurements are standard and appear sound.

The paper chooses questions resolved after each model’s knowledge cutoff and inputs are filtered to short, pre-resolution news summaries. This prevents potential leakage.

The paper evaluates a wide range of frontier models, including o3-pro, Deepseek-V3 and Claude-3.6-Sonnet.

**Weaknesses:**

I find the paper overall quite weak in its data collection and evaluation methodology.

On the data set, it only collections about 400 questions from one platform (i.e., Metaculus) with most questions from Jul–Dec 2024. It's unclear to me if the dataset is diverse enough. Compared with prior work like https://openreview.net/forum?id=FlcdW7NPRY, 400 is also quite a small sample size.

Regarding evaluation, the paper considers only 2 prompt styles and offers no attempt to optimize the overall prediction pipeline. For example, one could consider fine-tuning the model or optimizing the new retrieval system to improve the performance.  In addition, the paper’s own calibration plots show systematic miscalibration (Sec 5.5), but there’s no corrective post-processing or calibration method evaluated.

From an ML perspective, this paper provides no new technique in improving LLM for forecasting. It seems to be only an eval paper, and yet prior works like https://arxiv.org/abs/2409.19839 and https://futurex-ai.github.io/ are, in my opinion, stronger and more comprehensive. So it's not clear to me what is the novel contribution here.

Finally, from previous work, news retrieval can be unreliable when it comes to offering a date cutoff; see https://arxiv.org/abs/2506.00723. This can cause contamination. The paper would be stronger to test against this, as it applies a new pipeline called AskNews which to my knowledge has not been used widely in the literature.

**Questions:**

Any plan to release the dataset and model resolutions?

Is there a plan to continuously update the dataset and evaluate upcoming models?

Expert forecasts cover only subsets of (157 questions; 41 in the hold-out). Do you expect this is sufficient in estimating the model vs expert gap?

---

> ### Author Response · Authors · 2025-11-21
>
> We are thankful for your time and help in making the paper better. We were glad to hear that you thought the paper was well-written.
>
> ### 1. Low question numbers
>
> > On the data set, it only collections about 400 questions from one platform (i.e., Metaculus) with most questions from Jul–Dec 2024. It's unclear to me if the dataset is diverse enough. Compared with prior work like https://openreview.net/forum?id=FlcdW7NPRY, 400 is also quite a small sample size.
>
> Indeed the dataset is smaller than prior work, but it should be diverse enough for statistical significance. Additionally, I replicate the scores from previous papers for the overlapping models (ie. GPT-4o).
>
> ### 2. Only two prompt styles
> > Regarding evaluation, the paper considers only 2 prompt styles and offers no attempt to optimize the overall prediction pipeline. For example, one could consider fine-tuning the model or optimizing the new retrieval system to improve the performance. In addition, the paper’s own calibration plots show systematic miscalibration (Sec 5.5), but there’s no corrective post-processing or calibration method evaluated.
>
> These prompts were optimized early on during research and it has gone through several cycles of change, but we unfortunately did not rigorously record the differences.
>
> We did not add corrective post-processing because the models’ reported self-confidence level was a bi-modal distribution. In the prompt, we ask the models to provide a range of probabilities before giving a final answer, hoping to use this as a proxy for their confidence levels. The more extreme their forecasts, the more certain they report their confidence level (narrower range) in their answer, especially the worse the model. I’ve linked a scatterplot of [GPT-4o-mini](https://imgur.com/a/u9APPz4) and [o3](https://imgur.com/a/WtRJ8O7) (the best and the worst model by Brier score), with Brier score from each question graphed against the size of the range that they provided.
>
> ### 3. No new technique for improving LLMs in forecasting
> > From an ML perspective, this paper provides no new technique in improving LLM for forecasting. It seems to be only an eval paper, and yet prior works like https://arxiv.org/abs/2409.19839 and https://futurex-ai.github.io/ are, in my opinion, stronger and more comprehensive. So it's not clear to me what is the novel contribution here.
>
> This paper includes newer models and tests a different form of prediction/prompting (narrative) that others have [claimed that it works](https://arxiv.org/pdf/2404.07396)
>
> ### 4. Contamination
> > Finally, from previous work, news retrieval can be unreliable when it comes to offering a date cutoff; see https://arxiv.org/abs/2506.00723. This can cause contamination. The paper would be stronger to test against this, as it applies a new pipeline called AskNews which to my knowledge has not been used widely in the literature.
>
> It is true that news retrieval can be unreliable from date cutoffs, but the scores from the hold-out set, which contain questions from at least three months out from the models' stated knowledge cut-offs, do not differ much from the main dataset. Additionally, news articles were collected on the day of question opening (ie. not backcasting) for the hold-out set, so there would be no contamination.
>
>
> ### 3. Open-Source
> > Any plans to release the dataset and model resolutions?
> We do plan to release the dataset and model resolutions!
>
> ### 4. Expert forecast gap
> > Expert forecasts cover only subsets of (157 questions; 41 in the hold-out). Do you expect this is sufficient in estimating the model vs expert gap?
>
> We are less confident that the hold-out set is enough to evaluate the model vs. expert gap because of the small sample size, and thus higher noise levels; but both are subsets of the questions that the models forecasted on. We believe that over 100 questions should be enough for statistical significance, so the comparisons should hold for the main dataset. For the camera-ready version of the paper, we will add more statistical tests.
>
> Thank you again for taking the time to review the paper and providing feedback! Do the above address your concerns with the paper? If not, what further clarification or modifications could we make to improve it?

---

### Official Review · Reviewer_DzpJ · 2025-11-01

**Soundness:** 2
**Presentation:** 3
**Contribution:** 3
**Rating:** 0
**Confidence:** 4

**Summary:**

The paper presents an evaluation of LLMs against a select group of expert forecasters on real-world forecasting questions from Metaculus. Model prompts mimic human access to knowledge via inclusion of relevant news articles for the forecasting question in the prompt. Paper finds that LLM forecast are on par with general population, however underperform experts.

**Strengths:**

1. The paper contributes to measurement of LLM abilities in real-world reasoning and forecasting. The contributed dataset could potentially be of great value.

2. It is a timely problem to study.

3. Experiments consider most of the frontier closed source models.

**Weaknesses:**

1. The paper states various things without attribution/substantiation/citation. A couple of examples (among many) are as follows.
Lines 84 - 88, about mental model, ensemble of LLMs are better etc.
Line 105-106 statements like “ An LLM does better when fed news articles from AskNews over Perplexity”

2. I think the claim about preventing contamination as events haven’t occured yet based on LLM cutoff date is not entirely accurate. This is because the experiments are comparing LLMs performance with Metaculus experts, who may have opined about many of these events. The output is not contaminated as events haven’t happened but discussion around those may have happened. More broadly LLMs may have been trained on human analysis, discussion, and even prior predictions about these events (ifrom various other sources ncluding Metaculus itself) that existed before the knowledge cutoff. This is crucial to note because it forms the basis for experimental comparisons and results.

3. The paper compares LLMs to experts. However, experts are not defined appropriately. Paper mentions experts are the forecasters with  a track record of making good predictions. However, it is not defined what good means like paid experts, top 1% of predictors, or quantification. This makes it hard to understand how to interpret the results or their effects.

**Questions:**

1. Could you include description of experts and when were they identified relative to hold-out set curation?
2. Early on the paper, highlights that smaller datasets for similar tasks were quite smaller limiting its statistical power. Does it make sense to include in the  experiments statistical comparison to support statements like LLMs "significantly" underperform expert?
3. One of the prompts have words like “models are NEVER wrong”, which may introduce significant bias. How do results compare between direct and narrative prompts without such biased prompting?
4. Paper states that AskNews is better than Peplexity. Do you think it would be valuable to quantify how much of LLM’s performanc is attributable to quality of news itself? Would the conclusions of paper change if we use Perplexity or other search engines for news search?

---

> ### Author Response · Authors · 2025-11-21
>
> We are thankful for your time and help in making the paper better. We were glad to hear that you found the paper timely.
>
> ### 1. Missing description of experts
>
> > Could you include description of experts and when were they identified relative to hold-out set curation?
>
> These experts are defined as [pro-forecasters by Metaculus](https://www.metaculus.com/pro-forecasters/), where they have “at least 75 resolved questions and must have made predictions across multiple subject areas, with at least one year of experience making predictions.” These forecasters were identified before the tournament started: hold-out set forecasters were determined in September, and the main data set forecasters were identified in June.
>
> We will add a description of these forecasters and how they were chosen in the camera-ready version of the paper.
>
>
> ### 2. Overclaims without citation
>
> > The paper states various things without attribution/substantiation/citation. A couple of examples (among many) are as follows. Lines 84 - 88, about mental model, ensemble of LLMs are better etc. Line 105-106 statements like “ An LLM does better when fed news articles from AskNews over Perplexity”
>
> Thank you for pointing this out, we will add citations for these claims. For the points raised, ensemble of LLMs is better is from [Tetlock et al. 2024](https://arxiv.org/abs/2402.19379), and AskNews was worse than Perplexity was from individual experience and talking to various other forecasters in the [Metaculus AI forecasting tournament](https://www.metaculus.com/tournament/aibq1/). AskNews was better at filtering for publish dates than Perplexity, which would sometimes return out-dated articles and confuse the LLM (even when we pass the publication date of the article to the LLM).
>
> Additionally, we will also go through the paper for the camera-ready version to add citations to the claims or remove them if we do not have a citation.
>
> ### 3. Narrative prompt issues
> > One of the prompts have words like “models are NEVER wrong”, which may introduce significant bias. How do results compare between direct and narrative prompts without such biased prompting?
>
>  This likely does make the narrative prompts more biased, but the models oftentimes would output responses that are the reverse of what they think the actual outcome was because “ah it’s just fiction, we can’t always win at life” when we removed the emphasis that the models are never wrong.
>
> ### 4. AskNews vs. Perplexity
> > Paper states that AskNews is better than Peplexity. Do you think it would be valuable to quantify how much of LLM’s performance is attributable to quality of news itself? Would the conclusions of paper change if we use Perplexity or other search engines for news search?
>
> The conclusions of the paper may change if we used Perplexity over AskNews, but this is beyond the scope of this paper. We will add this to future research in the camera ready version. AskNews was better at returning articles within the scope of the time cutoffs. [Paleka et al. 2025](https://arxiv.org/abs/2506.00723) finds that “cutoffs are more of a guideline than a guarantee, and evidence suggests that models possess knowledge of some events beyond these dates,” so perhaps the results would not change significantly.
>
>
> ### 5. Contamination
> > the claim about preventing contamination as events haven’t occurred yet based on LLM cutoff date is not entirely accurate. This is because the experiments are comparing LLMs performance with Metaculus experts, who may have opined about many of these events
>
> As far as we could tell from spot-checking the news articles and doing Cntrl+F for Manifold, Metaculus, and Polymarket on the news articles, only one question included a result with odds from Polymarket.
>
> ### 6. Experiments for statistical comparisons
> > Early on the paper, highlights that smaller datasets for similar tasks were quite smaller limiting its statistical power. Does it make sense to include in the experiments statistical comparison to support statements like LLMs "significantly" underperform expert?
>
> Metaculus has a nice write-up on [noise in forecasting](https://www.metaculus.com/notebooks/14951/more-is-probably-more--forecasting-accuracy-and-number-of-forecasters-on-metaculus/), and the gist is that there's a lot of randomness with fewer than 30 questions. We will add statistical comparisons like t-tests in the camera-ready version.
>
> Thank you again for taking the time to review the paper and providing feedback! Do the above actions address your concerns with the paper? If not, what further clarification or modifications could we make to improve it?

---

### Official Review · Reviewer_F657 · 2025-11-02

**Soundness:** 3
**Presentation:** 2
**Contribution:** 3
**Rating:** 6
**Confidence:** 4

**Summary:**

This paper evaluates the forecasting ability of 12 state-of-the-art Large Language Models (LLMs), including frontier models from OpenAI (o3, o3-pro, GPT-4.1) and Anthropic (Claude 3.5/3.6). The author tests them on a new dataset of 464 real-world forecasting questions from Metaculus, with all events resolving after the models' knowledge cutoffs.

The evaluation includes a prospective and retrodictive setting, with a comparison of the results (the first of its kind, to the best of my knowledge). A narrative-based forecasting idea is also evaluated.

This paper is a bit of a curve ball. At first it seems like it will miss, but then once you notice its strengths it becomes quite promising. As such, I tentatively recommend acceptance (with the expectation that the authors will address comments raised by the reviewers). If the authors can address some of the framing issues, I would be willing to increase my score. If they can further strengthen the exploration of narrative-style forecasting, I think this paper could become quite strong and valuable for the community.

**Strengths:**

- The paper includes a prospective and retrodictive evaluation and compares them against each other. This has been sorely missing in the literature, but oddly I don't think the authors of the paper realize how much value this provides to the field. E.g., it partly resolves some of the key concerns highlighted by Paleka et al. (see weaknesses section for this citation). This comparison should be greatly emphasized. The fact that retrodictive and prospective evaluations give consistent results would be of interest to many people in the community.
- Figure 1 is compelling and should be placed at the top of the paper imo.
- The narrative prediction setting feels a bit wonky in the way it is currently presented (e.g., a discussion between Nate Silver and Tetlock seems weird; why not just have the model describe the future?). However, the core idea of getting models to describe the future in some form of prose rather than answer a binary question about the future (as is the focus of most forecasting work) is strong and deserves much more attention. If the authors can rework this to be more about the key distinction between structured prediction of the future rather than binary questions, I think the paper would be much stronger. As it stands, I think this is a bit too disconnected from the rest of the paper to strengthen it.

**Weaknesses:**

- Line 28 says "There are two types of forecasting: predicting the future based on a few datapoints or heuristics, or making predictions with a traditional machine-learning model". This doesn't seem like a natural taxonomy. Where does time series forecasting fit in? Also, forecasting like that done on Metaculus isn't made with few datapoints; enormous amounts of data (albeit unstructured) goes into those forecasts. To the best of my knowledge, the standard term for Metaculus-style forecasting is "judgmental forecasting".
- Only evaluating models on one news collection date is not ideal. How do models compare closer to the resolution date? This is very important, because humans may be more effective at incorporating new information, especially for more recent events after the pretraining cutoff of models.

Missing citations:
- "ForecastQA: A Question Answering Challenge for Event Forecasting with Temporal Text Data" by Jin et al. (a foundational paper in this area)
- "Forecasting Future World Events with Neural Networks" by Zou et al. (possibly the first paper to evaluate LLMs on Metaculus questions)
- "Pitfalls in Evaluating Language Model Forecasters" by Paleka et al. (crucial recent paper emphasizing the benefits of prospective evaluations of forecasting models)
- "LLM-as-a-Prophet: Understanding Predictive Intelligence with Prophet Arena" by Yang et al. (another recent prospective forecasting evaluation, along with Karger et al. (which is cited)).

Suggestions (not factored into score):
- RE Section 6.1, "All remaining errors my own": I would argue that any errors made by o3 and Claude in assisting with research are also attributable to the authors, much like a PI should take final responsibility for their students' errors.
- The paper says "This paper originally referred to expert forecasters as superforecasters, but I have since been informed that Superforecaster is a registered trademark of Good Judgment Inc." Oddly, "superforecaster" is frequently used in the rest of the paper. I don't think Good Judgment Inc would mind if the word "superforecaster" was used in a research paper, but I would recommend being consistent here.
- I feel like I'm going to lose this battle, but people really should be evaluating calibration error separately from Brier score (which conflates accuracy with calibration error). This is standard practice in non-forecasting treatments of calibration, but gets neglected by the forecasting community for some reason.

**Questions:**

See weaknesses

---

> ### Author Response · Authors · 2025-11-21
>
> We are thankful for your time and feedback, especially for suggesting a better framing. We were glad to hear that you found this paper interesting.
>
> ### 1. Framing
> We thank the reviewer for the constructive feedback, and we’ll emphasize the prospective vs. retrodictive evaluation in the camera-ready version of the paper, and move figure 1 to the top of the paper.
>
> We framed the narrative prediction for the model to inhabit the minds of top forecasters (ie. Nate Silver and Philip Tetlock), in attempts that they would make better forecasts. There is weak evidence that character prompting produces better results. Perhaps we would add a section to make the models to predict the future after reading a description generated by another LLM to make the comparison fairer to direct prediction.
>
> ### 2. Two types of forecasting
> > "There are two types of forecasting: predicting the future based on a few datapoints or heuristics, or making predictions with a traditional machine-learning model". This doesn't seem like a natural taxonomy. Where does time series forecasting fit in? Also, forecasting like that done on Metaculus isn't made with few datapoints; enormous amounts of data (albeit unstructured) goes into those forecasts. To the best of my knowledge, the standard term for Metaculus-style forecasting is "judgmental forecasting".
>
> We lumped time-series forecasting into traditional ML model forecasting, mostly because the common primary input is lots of data. We will adopt the term “judgmental forecasting” for the paper.
>
> ### 3. Evaluation dates
> > Only evaluating models on one news collection date is not ideal. How do models compare closer to the resolution date? This is very important, because humans may be more effective at incorporating new information, especially for more recent events after the pretraining cutoff of models.
>
> The average time between question release to question resolution is 38 days; the maximum was 85 days; and the minimum was 1 day. How models do on questions that are further out/closer to resolution date would be a good future line of research; our suspicion would be that models do better on short-term questions. (Halawi et al. 2024)[https://arxiv.org/abs/2402.18563]'s average was 42 days, going up to 70 days.
>
> ### 4. Calibration vs. accuracy
> > I feel like I'm going to lose this battle, but people really should be evaluating calibration error separately from Brier score (which conflates accuracy with calibration error). This is standard practice in non-forecasting treatments of calibration, but gets neglected by the forecasting community for some reason.
>
> We do show a calibration plots, but would be happy to add expected calibration error in the camera-ready version.
>
> We'll also go through the paper thoroughly and remove all mentions of superforecasters; this was an oversight (they sent us an email about how it was trademarked and thus we cannot use the term to refer to Metaculus pro forecasters unfortunately).
>
> Thank you for bringing up the missing citations; we will make sure to include them.
>
> > RE Section 6.1, "All remaining errors my own": I would argue that any errors made by o3 and Claude in assisting with research are also attributable to the authors, much like a PI should take final responsibility for their students' errors.
>
> I thought this is what "all remaining errors my own" means? ie. any errors in the paper is the responsibility of the authors, not the RAs or AIs.
>
> Thank you again for taking the time to review the paper and providing feedback! Do the above actions address your concerns with the paper? If not, what further clarification or modifications could we make to improve it?

---

### Meta-Review · Area_Chair_ypyM · 2025-12-24

**Summary:**

This paper evaluates the forecasting capabilities of 12 state-of-the-art Large Language Models (including frontier models like o3 and DeepSeek-V3) on a dataset of 464 questions from Metaculus. The study utilizes a hold-out set where news articles were collected in real-time to prevent data leakage. The results suggest that while LLMs outperform the human crowd, they continue to underperform expert forecasters , and that narrative-style prompting generally yields worse calibration than direct prompting.

The AC recognizes that the reviewers’ feedback is mixed. While the AC appreciates the timely evaluation of frontier models and the rigorous attempt to prevent contamination via the hold-out set, the decision recommended by the AC, is that the paper is not yet ready for publication at ICLR, primarily due to the limited methodological novelty and scope.

**Reviewer Concerns:**

**Resolved Concerns:**
- Data Contamination (Reviewer DzpJ): The reviewer strongly objected based on the belief that experts might discuss events online, leaking future knowledge to models. The author successfully clarified that for the hold-out set, news was collected on the day of question opening, strictly preventing future information leakage.
- Definitions and Citations (Reviewer DzpJ, F657): The authors provided the missing definitions for "experts" (Metaculus Pro Forecasters) and agreed to add missing citations for claims regarding ensemble performance.
- Framing and Terminology (Reviewer F657): The author agreed to fix the "natural taxonomy" of forecasting and remove the trademarked term "Superforecaster," addressing the reviewer's specific framing concerns.

**Outstanding Concerns:**
- Methodological Novelty (Reviewer ih4C): The rebuttal did not fundamentally alter the fact that this is a pure evaluation paper without a novel algorithmic contribution or optimization of the prediction pipeline. The "narrative prediction" method, which could have been a novel contribution, was found to perform worse than direct prediction and was described as "wonky" by Reviewer F657.
- Dataset Scale (Reviewer ih4C): While the author argued that ~400 questions are sufficient for statistical significance in a live setting, the dataset remains significantly smaller than comparable benchmarks (e.g., ForecastBench), limiting the robustness of the conclusions regarding model vs. expert gaps.
- Presentation Polish (Reviewer nVva): Although the author promised a rewrite , the current state of the manuscript ("poorly written project report" ) requires a major overhaul that exceeds the scope of a standard camera-ready revision.

**Reviewer Scores:**

- Reviewer F657: (6 to 7). The reviewer was already positive ("tentatively recommend acceptance" ), and the author accepted all suggestions.
- Reviewer DzpJ: (0 to 3). The "Strong Reject" was based on a technical misunderstanding about data leakage. With the hold-out set methodology clarified, the score should be improved.
- Reviewer ih4C: (2 to 3). Likely still skeptical about the sample size, but may acknowledge the value of the frontier model results.
- Reviewer nVva: (2 to 3). There were no substantial concerns other than styling and writing issues.

---

### Decision · Program_Chairs · 2026-01-26

Reject